# Dimensions of Community Assets for Health. A Systematised Review and Meta-Synthesis

**DOI:** 10.3390/ijerph18115758

**Published:** 2021-05-27

**Authors:** Pablo Alberto Sáinz-Ruiz, Javier Sanz-Valero, Vicente Gea-Caballero, Pedro Melo, Tam H. Nguyen, Juan Daniel Suárez-Máximo, José Ramón Martínez-Riera

**Affiliations:** 1Department of Community Nursing, Preventive Medicine, Public Health and History of Science, University of Alicante, 03080 Alicante, Spain; pabloalberto.sainz@gmail.com (P.A.S.-R.); jr.martinez@ua.es (J.R.M.-R.); 2Adscript Center of Universidad de Valencia, Research Group GREIACC, Health Research Institute La Fe, Nursing School La Fe, Avda. Fernando Abril Martorell, 106. Pabellón docente Torre H, Hospital La Fe, 46026 Valencia, Spain; 3Centre for Interdisciplinary Research in Health, Institute of Health Sciences, School of Nursing (Porto), Universidade Católica Portuguesa, 4169-005 Porto, Portugal; pmelo@porto.ucp.pt; 4Center for Health Technology and Services Research, NursID Project, 4200-450 Porto, Portugal; 5William F. Connell School of Nursing, Boston College, Newton, MA 02467, USA; tam.nguyen@bc.edu; 6Titular de la Oficina Estatal de Educación de la Asociación Mexicana de Estudiantes de Enfermería, División Puebla, Teziutlán 73800, Mexico; juan.suarezma@alumno.buap.mx

**Keywords:** health assets, salutogenesis, dimensions, categorical analysis

## Abstract

Since Aaron Antonovsky’s salutogenesis theory and Morgan and Ziglio’s health assets model were first proposed, there has been a growing concern to define the resources available to the individual and the community to maintain or improve health and well-being. The aim of the present study was to identify the dimensions that characterise community assets for health. To this end, we conducted a systematised review with a meta-synthesis and content analysis of research or projects involving asset mapping in the community. Articles that met our eligibility criteria were: (1) based on the salutogenic approach and (2) described an assets mapping process and among their results, explained what, how and why particular community assets for health had been selected. The search included primary studies in the published and grey literature which were selected from websites and electronic databases (Web of Science, MEDLINE, Scopus, EBSCOhost, Dialnet, SciELO). Of the 607 records examined by a single reviewer, 34 were included in the content analysis and 14 in the qualitative synthesis. Using an inductive process, we identified 14 dimensions with 24 categories, for which in-depth literature reviews were then carried out to define specific indicators and items. These dimensions were: utility, intention, previous use, accessibility (“circumstances–opportunity–affordability”), proximity-walkability, connectivity, intelligibility (visibility, transparency), identity (uniqueness, appropriability, attachment), design (configuration, functionality, comfort), safety (objective/subjective), diversity, the dimension of public and private, and sustainability (which includes maintenance, profitability or economic sustainability, environmental sustainability, centrality-participation and equity-inclusiveness).

## 1. Introduction

Numerous authors in the fields of psychology and the social sciences have attempted to define the resources available to the individual and the community to maintain or improve health and well-being. These have included Aaron Antonovsky, whose theory of salutogenesis defines the Sense of Coherence and General Resilience Resources (GRRs) [1,2], and Kretzmann and McKnight [3], whose Asset Based Community Development (ABCD) model transfers the concept of *assets* to the community. Eriksson and Lindström’s [4] “salutogenic umbrella” covers many of the concepts and theories that share a positive approach to explaining people’s health and quality of life, in contrast to the traditional biomedical approach which focuses on deficits, the treatment of diseases and prevention against risk factors.

In general terms, these can all be referred to as what Morgan and Ziglio [5] have called “health assets”, which range from intra-personal assets, such as Antonovsky’s sense of coherence, Kobasa’s hardiness and Werner’s resilience, to inter-personal assets such as Putnam’s social capital and Bourdieu’s cultural capital. These are all protective health factors and appear implicitly in multiple proposals, such as Scales and Leffert’s [6] synthesis of the literature related to the Search Institute’s “developmental assets”, aimed at guiding health promotion strategies for young people. 

Antonovsky defined GRRs as any characteristic of a person, group or environment that facilitates effective stress management, and they can be genetic, biological, physical, material, cognitive, emotional, attitudinal, relational, sociocultural, spiritual or psychosocial in nature [1,2,7]. In their ABCD model focusing on local assets and oriented towards “relationship driven”, Kretzmann and McKnight [3] emphasise the importance of the role of the community in identifying individual and collective capacities and talents and environmental strengths or resources available in the context [8]. Morgan and Ziglio defined health assets for the first time as “any factor (or resource), which enhances the ability of individuals, groups, communities, populations, social systems and/or institutions to maintain and sustain health and well-being and to help to reduce health inequities” [5] (p.18). In their asset model, they advocate using Kretzmann and McKnight’s method as a practical approach to public health based on Antonovsky’s salutogenic orientation. It goes far beyond intra-personal assets to encompass practically anything that a community identifies as its own that can potentially benefit coexistence, development and health. Other authors such as Rotergard et al. [9] have related the antecedents of assets to the determinants of health, thus broadening the concept to considerations of health inequities such as socio-economic conditions, inclusiveness or accessibility.

All of the above renders it difficult to frame the concept of health assets within an operational definition that can be used to plan health promotion strategies or implement community interventions. Some difficulty has also been observed in reaching consensus on which assets might have the greatest influence on community health and why. This question has been highlighted in the literature, particularly in descriptive studies of community interventions, such as the research by Aviñó [10] and Jakes et al. [11]. Jakes et al. contend that there is a need to further examine “when a resource becomes a GRR” [8] (p. 167) and stress the importance of developing appropriate indicators and explore the values underlying decisions. 

The systematised review presented here was largely motivated by the need to answer some of the following questions: What differentiates a *community asset for health* from other resources? Are all resources potential *community assets for health*? The answers to these questions will contribute to the long-sought goal of researchers, practitioners and policy makers to define suitable methods for measuring and evaluating asset-based approaches [12].

According to Stokols et al. [13], health promotion campaigns should be prioritised according to health problems and social and physical environments that are directly related to particular needs, strategically matching resources to pressing concerns. Assets gain meaning in the context of needs, while the latter become meaningful in the quest for assets [14]. There is also widespread interest in determining synergies between the salutogenic approach and the deficit model to leverage the complementarity of both, recognising the dialectic links between needs and assets, or between protective factors and risk factors, echoing Antonovsky when he referred to the experience of health as a ease/dis-ease continuum [14,15,16,17]. Van Kamp et al. [18] have advocated developing interdisciplinary, intersectorial tools for application in real-life policy and decision-making activities.

### Review Question

The aim of this study was to identify the dimensions that characterise a community asset for health in order to design a tool that facilitates identification and assessment of these assets. To do so, we will systematically review all the evidence to describe the “universal” characteristics that are socially considered the defining qualities of physical and community resources to perceive them as health assets.

## 2. Materials and Methods

### 2.1. Procedure and Framework

This review and content analysis was carried out as part of a doctoral research project entitled “Identification and Assessment of Assets for Health: Epistemological Analysis and Measurement Model”, at the University of Alicante (Spain).

In order to identify the dimensions that are “universally” considered by observers in asset mapping processes, the first step was to conduct a systematised review and meta-synthesis of articles and grey literature that reported mapping research or projects, described the work process and gave the results obtained as an inventory of healthy resources/assets, e.g., studies that answered the questions of what, how and why particular community assets for health were selected.

Once a taxonomy of dimensions had been defined, following an extended period of searching for and reading research in different fields of study, we specified the criteria that defined them and the items that enabled their analysis in order “to design a tool that facilitated the assessment and weighting of community assets for health”. The theoretical model and proposed tool were presented to a panel of experts, using the Delphi method to analyse and debate the crucial issues and modify those aspects on which consensus was reached. 

Assuming that several, if not most, of the studies would be case studies or constitute exploratory primary research of a qualitative nature [12,19,20], the literature search was guided by a SPICE question [21] that captured the concepts of interest: what are community resources for health and why are they identified as such by the general population when asset mapping processes are carried out in a territory? Our research question did not seek to compare interventions, but to identify all those actions “in and with” the community, regardless of territory, time, participant sampling criteria or the group participants represented.

### 2.2. Search Strategy

Based on the structure of the research question, we defined the search terms, the document eligibility criteria and the search strategy (i.e., databases, journals and search engines to consult).

The keywords and phrases used in the literature search were selected so as to yield the highest possible sensitivity and specificity, despite the lack of MeSH and DeCS terms on salutogenesis and health assets. The keywords employed were: health asset*; health resourc*; communit* asset*; build* health asset*; asset* based; and, map* asset* and their corresponding translations into Spanish.

The results of this systematised literature search are summarised in a flow diagram adapted from Moher et al. [22] and shown in Figure 1, providing transparency as regards the method employed.

### 2.3. Eligibility Criteria

Studies were included if they met the following eligibility criteria.

#### 2.3.1. Theoretical Approach or Framework

The eligibility criteria included studies that were based on the salutogenic approach and explored the principles of asset-based thinking in the context of health. This encompassed studies aimed at identifying assets (personal, collective and physical) or studies that adopted a mixed approach, identifying community strengths but also detecting needs or problems.

#### 2.3.2. Types of Study Design 

All types of study were eligible for inclusion, albeit most appeared to be of a qualitative nature. Articles such as commentaries, meeting papers, editorials or opinion statements were not considered for inclusion.

#### 2.3.3. Types of Outcome 

Studies whose results identified physical or community assets, beyond internal personal assets, were selected in the form of a synthesis or asset map. In addition, the studies had to contain a description of the qualities of the resources identified, detailing “what” assets had been identified and explaining “why” they had been selected, answering the question: why did the participants in each study chose some resources as assets and not others?

### 2.4. Information Sources

The search strategy for this review aimed to identify primary studies, grey literature, and reviews in electronic databases and by manual searching. We searched the following electronic databases for primary studies from database inception up to the search date (last search August 2020). This search was applied to:-Web of Science-Scopus-MEDLINE (via PubMed)-EBSCO host-Cochrane Public Health-Dialnet-SciELO

No restriction date was used. No restriction language was used.

#### Other Sources 

Journals, reference lists of included studies and previous scoping reviews related to salutogenesis and health assets were manually searched for additional studies. We also used other grey literature search engines such as Springer Link and TESEO or DART for doctoral theses. The systematised review was complemented by a comprehensive search of Internet resources to identify grey literature on the subject, including websites specific to the area of research, such as the Center on Salutogenesis, and web resources sharing experiences of mapping, such as the “Red de Actividades Comunitarias” (community assets network) (Spain). These resources enabled us to identify, for example, the experience of “Mapping Puerta del Ángel” in Madrid (Spain) [22].

### 2.5. Selection of Sources of Evidence

Of the total of 675 records screened, duplicate records were removed, and then one reviewer screened the titles and abstracts. In the case of studies whose appropriateness was unclear, the full text was screened. All articles included after this stage were read in full according to the eligibility criteria. It is important to note that in accordance with Grant and Brooth [23], this was a systematised rather than a systematic review because the documents were assessed by a single reviewer and were not peer-reviewed. In order to reduce selection bias and ensure intra-observer reproducibility, the review was conducted in two stages several months apart, between late 2018 and October 2019, with a final review in August 2020. 

The following data were extracted from the included studies:(1)Publication information (title, author and date of publication, local place and country);(2)Study characteristics (design, theoretical and methodological approach, population of interest, objectives of the studies);(3)Health asset characteristics: type of resources (personal, inter-personal and community assets) and dimensions or factors that defined the assets identified.

### 2.6. Quality Appraisal of Included Studies 

The methodological quality of the selected studies was assessed during data extraction.

The studies were initially assessed using the Equator Network COREQ checklist available at https://www.equator-network.org accessed on 20 March 2018, although the authors of this guide indicate that it was developed “to promote explicit and comprehensive reporting of qualitative studies (interviews and focus groups)” [24] (p. 356) and several of the studies included in the review consisted of action research with mapping workshops. Consequently, we also took into account the recommendations given in the SRQR guide and the EPICURE analysis proposed by Stige, Malterud and Midtgarden [25]. Critical appraisal sheets were then used to qualitatively assess the methodological rigour of each study. We considered a study to be of good quality (++) when all criteria were met (detailed description of the study design: context, sample and method; ethics; and the quality of analysis and results), of fair quality (+) when most of the criteria were met and of poor quality (–) when most of the criteria were not met (low quality criteria or no appropriate results). 

### 2.7. Synthesis, Content Analysis and Categorisation

The documents obtained as a result were examined by meta-synthesis and are summarised individually and synthetically in Table 1, giving information from each study on: location and context, study design, population and methods used, results and findings obtained (type of health assets mapped: “only personal health assets”, “mixed health assets”, “only community assets for health”; where the study gave an inventory of the identified assets and/or the reasons or characteristics) and a quality assessment. 

We then performed an inductive process of categorisation and taxonomy of the results of the primary studies. Data extracted from the selected studies were coded by content analysis, grouped by categories and reported in concept diagrams. Some of the dimensions that emerged from the inductive process were “imported concepts” from the primary studies, while others were “in vivo” concepts, conceived by the researcher. The ATLAS.ti software version 7.5 was used throughout the procedure.

A list of “universal dimensions” was inferred which we judged to be the qualities valued by observers and/or users of community resources. 

Then, from the taxonomy of concepts that emerged from the inductive analysis, in-depth literature reviews of each of the categories were conducted to identify specific indicators and items. Examples of the keywords used include: accesib*, affordab*, attitude*, availab*, design, util*, built environment*, “neighbourhood features”, “neighbourhood attributes”, sustainab*, safety/security, visible/visibility, walkabl*. 

## 3. Results

### 3.1. Study Characteristics of Included Studies

The main characteristics and quality assessment of the included studies are summarised in Table 1. We included a total of 14 reports, which were published between 2010 and 2019. 

Twelve were original articles, most of which used a purely qualitative approach, as did Aviñó’s doctoral thesis [10], which studied two community development interventions using the asset-based approach, and a report describing an assessment intervention in an African community [37].

An analysis of the documents finally included in the systematised review showed that a large number of the studies had been conducted in the United States (29%), Spain (21%) and England (14%). Nevertheless, also included was a study by Jabeen [32] carried out in Bangladesh, the Railton Foundation’s community intervention in South Africa carried out by Lazarus et al. [37] and studies by Den Broeder et al. [29] and O’Connor et al. [26], conducted in the Netherlands and Australia, respectively. On average, the studies were performed around 2011 and half of them had been published before 2015. 

As regards methodology, 10 of the 14 studies (71%) adopted Kurt Lewin’s participatory action research approach, most of which (79%) used the CBPR research model [27,30,31,35,37], while Aviñó [10], Greetham et al. [36] and Matthiesen et al. [34] employed the ABCD method proposed by Kretzmann and McKnight [3], and others used specific variants such as the CHAMP method for associations, which was employed by the Railton Foundation and the University of Stellenbosch in Lazarus et al. [37]. 

The remaining studies also adopted a descriptive qualitative approach, but instead of action research techniques they employed qualitative methods such as semi-structured interviews, surveys with open-ended questions or focus/nominal groups, and the data were examined exclusively by the researchers. In general, the studies employed the “asset-based approach” (AB) or were based on Aaron Antonovsky’s salutogenesis theory, as in the case of the study by Sánchez-Casado et al. [28]. Others, such as the studies by Pérez-Wilson et al. [33] and Jakes et al. [11], were based on the Health Asset (HA) model. 

In this type of research, where the study participants are the subject of analysis (e.g., their behaviours, their discourses), even if they sometimes take an active part in the research, it is important for the researcher to take ethical considerations into account. However, these were only explicitly mentioned in six of the 14 studies reviewed (43%) [29,30,31,34,35,37]. However, since the report by Greetham et al. [36] was a pilot study funded by the National Health Service, it can be assumed that it observed ethical principles.

Although sample size is not a criterion of quality in qualitative studies, the process of identifying participants is. An analysis of the theoretical and practical approaches on which these studies were based showed that participant sampling was mainly situational (43%), followed by intentional sampling according to a situation chosen by the researchers (36%) and convenience sampling according to accessibility (14%). In some cases, participants were recruited by means of “snowball” or “cluster” sampling [33], and in others, according to relevance [28] or to maximum variation, i.e., seeking to include all possible aspects of the phenomenon in the sample. 

The study population and samples differed between studies. Thus, some studies focused on a specific group, as was the case of the research by Pérez-Wilson et al. [33] and DyckFehderau et al. [35], which sought to determine the perceptions of children or adolescents, or on groups characterised by a particular phenomenon, such as having diabetes [30] or end-of-life problems [34]. Others, such as the study by Cutts et al. [31], the thesis by Aviñó [10] and the research by O’Connor et al. [26] sought to compare groups from different situational contexts or generations (young people-older adults). With the exception of Cutts et al. [31] and Pérez-Wilson [33], none of the studies provided information on the age or sex of the study participants (*n* = 12; 86%), and only three indicated the participants’ role or profession [28,29,34].

### 3.2. “Universal” Characteristics of Community Assets for Health

The content analysis and categorisation process yielded the universal dimensions of community assets for health shown in Table 2, where it can be seen that three dimensions in particular (“walkability”, “safety” and “participation”) presented strong links with others. These conceptual links, also understood as concurrences, only explained the relationships between the categories inferred from the content analysis. Meanwhile, the most frequently codified concepts were “affordability”, “maintenance-care”, “naturalness” and again “safety”.

We also categorised the key needs that participants frequently referred to in their personal orientation when assigning a value to a resource. From highest to lowest frequency of occurrence in the texts, the most frequently coded needs were: health (need)–activity (need)–fun/leisure (need)–role and relationships (need).

After conducting in-depth literature reviews for each of the concepts, 14 dimensions were identified, encompassing 24 categories and 145 items (Table 3).

#### 3.2.1. Utility

Underlying the interpretation of a resource as meaningful for the individual or community was the condition of “utility”, and these concepts were directly related, i.e., a resource was meaningful when it was useful. However, utility is not an inherent or intrinsic dimension of a resource, but is subjective in nature and varies according to the needs, values and culture of individuals and communities. According to Abraham Maslow’s model of the hierarchy of human needs or motivations [38], awareness of a need will generate a desire to meet it. 

Among the factors external to a resource, utility is the first dimension that can be assessed, in terms of the capacity of the resource to meet the needs or demands of the community [39].

#### 3.2.2. Intention

According to Azjen [40], “attitude” precedes behavioural “intention”. The Theory of Planned Behaviour (TPB) provides an insight into the interaction of dimensions that contribute to the behavioural process of any individual, particularly as regards the decision chain involved in using a resource: “previous use” and “perceived accessibility” [41].

*“[…] Among the identifed internal assets were well-being, happiness, a positive attitude towards health, self-confdence, acceptance, respect, self-esteem and the ability to handle difculties and challenges”* [42] (p. 258)

According to the Theory of Reasoned Action, an extension of the TPB, intention is the primary motivator of behaviour, and is understood as a function of two independent constructs: Subjective Norm and Attitudes [40]. As can be seen in Van Kamp et al. [18], environmental quality is a complex issue involving subjective perceptions, attitudes and values that vary between groups and individuals. Consequently, the more positive an attitude and the more consistent it is with subjective norms, the stronger the intention will be, thus motivating a greater effort to carry out an activity or behaviour. However, actual behaviour may differ from intended behaviour, because individuals do not have sufficient control over all the variables that condition it. They will be more motivated when they perceive that their behaviour can lead to success. This perception of control will depend on the degree of difficulty or the belief that this can be overcome and on the perception of internal control versus the influence of external factors [43,44].

#### 3.2.3. Previous Use

Our content analysis revealed several perceptions of the participants in the studies analysed regarding the influence of their previous use of a resource on whether or not they perceived it as a health asset. A shortcoming of the Theory of Planned Behaviour is that it does not consider the influence of past behaviour when predicting future behaviour. Several studies also indicated that inclusion of a “previous use” behavioural variable in the TPB theory improved the predictive capacity of the model [43,44].

#### 3.2.4. Accesibility (Perceived): Affordability, Proximity, Walkability, Connectivity and Legibility

While the vast majority of the studies referred to accessibility as synonymous with proximity or availability, such an interpretation is neither conceptually nor practically correct [45]. Numerous studies revealed inconsistencies between measures according to quantitative standards and the subjective interpretation of accessibility [41].

To speak of perceived accessibility implies considering it as a property of the individual with respect to the resource and the environment. As Pirie (1979) noted, “accessibility is always created and is not just something to be had by virtue of one’s locale” [46] (p. 307). 

The inductive analysis enabled us to organise accessibility according to the following broad categories: affordability, proximity, walkability, connectivity and legibility.

The concept of proximity is frequently mentioned in the literature and concrete proposals have been made for its measurement. However, several studies have indicated a discrepancy between actual physical distance and the known distance [8,47]. Consequently, numerous researchers prefer to set proximity thresholds in terms of time, ranging from 5 min (400 m) to 30 min (2400 m) as the ideal distance for walking [48,49], or to measure proximity according to territorial delimitations such as communities or districts (1200–1600 m) [48,50,51,52,53,54,55,56], smaller areas such as neighbourhoods, not exceeding 10 min walking time [49,57,58,59,60] or according to “activity areas” [61]. 

*“[…] the long distance, as well as a lack of information are the primary reasons why some societies are excluded from getting access to health services”* [62] (p. 6).

In addition to the proximity of a resource, it is necessary to consider how the individual accesses it. The concepts of “connectivity” by means of transport and “walkability” in the sense of sufficient environmental quality to move from one resource to another on foot appeared frequently in the content analysis. Walkabilty is also often mentioned in the literature in relation to factors such as the aesthetics of a route, the diversity of land uses that attract the walker’s attention and the condition or maintenance of the road and lighting.

It is often related to the “three Ds” (density, design and diversity) proposed by Leslie in the *walkability index* [53,55,56,63,64,65]. It has been found that the perceived design or configuration of a space is an influential factor in people’s attitudes towards walkability. Directness, route integrity and mixed land uses influence this perception [53,63,66,67,68]. 

Lack of transport is an issue that resurfaced constantly in the studies analysed and in the literature in general. “Connectivity” was categorised as all statements referring to the ease or otherwise of access to a resource via public transport or private car [48,69,70,71,72]. Millward et al. [68] advocate the frequent use of 400 m distance thresholds when planning public transport routes and stops.


*“[…] Transport and connectivity was a dimension that the professionals considered health enhancing. The professionals regarded the traffic infrastructure”*
[29] (p. 7)


*“Lack of accessible and reliable public transportation may increase the need for financial resources, to have extra time or having to roll long distances to get to the asset of interest”*
[73] (p. 7).


*“Poor transport and communication infrastructure in many rural communities are noted to exclude many from having adequate access to healthcare”*
[62] (p. 6).

Other equally important categories that influence perceptions of a resource as accessible are the factors of “legibility” and “affordability”. Legibility refers to the visibility of a resource in a territory in relation to its physical transparency and the information on or dissemination of its services. 


*“Accessible information and resources shared between organisations in the community […] Available in print and on line”*
[34] (p. 311)

Meanwhile, the category of affordability includes factors related to individual circumstances or particularities (e.g., physical, cognitive, mental) and the idea of economic accessibility and time opportunities [13,31,45,51,74]. Again, according to the TPB, when there is an opportunity to act, intention is converted into behaviour [44].


*“[…] large food purchases were sometimes made at supermarkets outside of town to access a wider range of foods at potentially lower prices”*
[75] (p. 3).

#### 3.2.5. Identity

“Identity” refers to an individual’s feeling of attachment to physical places or spaces, which generates self-esteem and a unique bond. A very clear example of this is religious resources, which satisfy the basic need for “values and beliefs”: 

*“Churches give a sense of hope. People respect church...”* [31] (p. 11). *“Several participants described their active involvement in religious traditions, but they mentioned the inconvenience of not having a church where they could hold mass”* [76] (p. 8).

Social identity and place identity are closely related [77,78]. For a space or resource to be considered a symbolic space, it must be socially perceived as prototypical, i.e., paradigmatic or representative of the category on which social identity is based.

In our content analysis, we found references to the dimension of identity in several closely related concepts. The last taxonomy in this dimension was organised according to the 20-item Urban Identity Scale by Lalli and Thomas [79]. This includes singularity, appropriability and attachment. “Singularity” is the quality of being extraordinary, rare, excellent or out of the ordinary in comparison with similar elements, perhaps because of its historical-cultural particularity [10,70,80,81,82]. “Appropriability” comes from the concept of appropriation of place, when an individual interacts with the environment and appropriates its social meanings [83]. However, people sometimes value the existence of some resources, such as parks, even when they do not use them, “just for being there” or “just having them around”, for their indirect or future benefits or for third parties [41,84]. “Attachment” may be physical, with an individual feeling part of a place, or social, through a feeling of durability and immutability [79,85].

#### 3.2.6. Design

Design encompasses the ideas of configuration, functionality and comfort, in parallel with the principles proposed by Vitruvius—*venustas, utilitas, firmitas*— and according to the items organised using the Design Quality Indicator scale [86].

Configuration refers to the factors of composition and organisation of a resource and of the space, form and materials, but also to character and beauty (aesthetics), such as the “green roofs” widely referred to in the studies reviewed [45,87,88,89]. Physical attributes are given most importance in quality analysis, whereas organisational and social characteristics receive relatively little attention [39].

The “functionality” of resources also receives attention, in terms of the capacity of a resource to perform various functions, or the multifunctionality of an asset that simultaneously serves several fundamental needs: “*Schools should have more functions to include the community*” [37] (p. 65).

Comfort is a conceptual category that includes all references to friendliness and pleasantness, such as hygrothermal comfort, acoustic comfort, air quality comfort or visual comfort [90].

#### 3.2.7. Safety (Perceived and Objective)

Our content analysis revealed that the concept of “safety” is shaped by subjective perception—whether individual or collective—and by objective measurements, albeit these are usually interpreted subjectivity. While this is the case in most of the dimensions mentioned above, in the case of “safety” it is more common to find discourses that address objective issues such as crime rates, human presence or infrastructure deficiencies: 


*“Safety concerns ranged from environmental hazards and limitations of the physical environment (e.g., few parks and green areas; poor maintenance of existing parks) to neighborhood threats (e.g., thefts, gangs, vandalism) and domestic violence.”*
[91] (p. 5).


*“[…] adolescents felt afraid to use those green spaces because ofthe presence of gang activity and drug sales. Youth expressed the need for more safe recreation facilities that are appropriate for adolescents and equipped…”*
[76] (p. 8).

The data show that people who live in safe and friendly environments are more active and make use of resources [82,92].

A number of studies have associated perceptions of safety with high levels of participation, for example in an association or in an open space [31]. Participating in and engaging with a resource increases an individual’s confidence, and this in turn improves perceptions of safety.

#### 3.2.8. Diversity

The dimension of diversity was created to capture ideas about the range of health assets in a territory (external perspective) and the “variety of supply” or of facilities for the same function (internal perspective), and is related to the ideas of availability and affordability: “*large food purchases were sometimes made at supermarkets outside of town to access a wider range of foods at potentially lower prices*” [75] (p. 3).

The greater the internal diversity —for example, the diversity of fruit or vegetables in a grocery shop, of vegetation in a park or of activities in a cultural association or sports centre—the more valuable a health asset is considered to be. From the perspective of context, diversity refers to a community or territory’s endowment in terms of number and variety of resources that perform the same function. Thus, the lower the endowment of a resource in a territory, the higher the value of that resource because of its scarcity. Shannon’s diversity formula yields objective measures of diversity, but other specific scales also exist depending on the resource analysed, such as the *Nutrition Environment Measures Survey* (NEMS) for restaurants or convenience stores [93].

#### 3.2.9. Public

The idea of “public” encompasses the perceptions of identity that users have of resources that are not intended for proprietary or exclusive use, whether for financial or other reasons: “*If you have to pay, then it’s not really a [community]*” [91] (p. 6).

However, we also decided to include participants’ references to private resources in this dimension. 

#### 3.2.10. Sustainability

The sustainability dimension encompasses a wide range of concepts, such as the resilience of an asset over time, its intersectorality or community participation, and other values such as environmental sustainability and its role in reducing social inequalities. 

At the United Nations Summit on Environment and Development in Rio de Janeiro (1992), the concept of “sustainable development” was defined as a system that seeks a balance between economic, social and environmental development processes. 

Thus, an asset should be durable, socially profitable, environmentally friendly and a guarantor of greater social capital and equity [94]. 

The durability of a resource will depend on the physical condition of the site (maintenance) and on human activity (leadership, human team), in so far as these affect its functioning or organisation:


*“[…] the capacity of a system to maintain structure and function when faced with shocks and change” “[…] resilience-building in social-ecological systems are structured scenarios and active adaptive management”*
[95] (p. 49).

The concepts of inclusion and participation are also related to the sustainability of a resource from a social perspective, and both refer to a resource in its territory (intersectorality) and to the community in the functioning and durability of the resource [96].

Intersectorality is linked to the construct of sustainability, and includes concepts such as adaptability, stability and transformability, in the sense of innovation: “*Forge links to existing structures/organisations for on-going sustainability*” [34] (p. 311). The cross-sectoral activity of a resource, or its degree of participation in the community, can be understood as the number of actors who come into direct contact with its activity. The use of quantitative network analysis methods has been proposed to understand the “centrality” of the resource in a territory [97].

The concept of “socialisation” is closely related to participation, although they do not always go hand in hand. This category can be defined as the social capital of resources, which fosters social relations and frequently refers to resources such as parks or cafés: “*[…] healthcare users are proud of the market and point at the cafeterias as hubs of socialisation*” [98] (p. 9).

Lastly, “inclusiveness” and empowerment in the functioning of a resource allow for the creation of spaces of exchange that help to promote equity, where everyone is treated respectfully and on an equal footing according to their capacities and needs [11,99].

## 4. Discussion

This study provides a comprehensive systematised review of the literature on health asset mapping and for the first time differentiates the dimensions and concepts that “universally” define a resource as a community asset for health.

Asset mapping is only the first stage of a process aimed at connecting and mobilising assets [36], and as can be seen in most practical guides [36,100,101]. The next step is to prioritise/weight the assets identified before undertaking strategic actions. This stage is often particularly complex because in many cases it is ambiguous for citizens and highly variable depending on who performs it.

This may explain why our systematised review and literature searches revealed that many of the studies describing asset mapping experiences used this procedure as a means to engage local communities in the process, presenting their results as an inventory of intra-personal strengths or of community assets for health; however, these failed to provide an overview of the territory’s health, or were not used to plan possible community strategies or health promotion interventions. This finding has been reported by other authors such as Morgan and Ziglio [5], Pons-Vigués et al. [98] and more recently Van Bortel et al. [102]. Our results show that there has generally been an attempt to map individual, relational or environmental assets, paying attention to protective or health-promoting factors rather than asking the question “why”? Much effort has been invested in identifying these assets (at personal, societal or community level, and in different contexts), rather than in determining the motives that drive individuals or the community to strengthen some resources, or the common qualities that explain why more than one individual perceives a resource as a community asset for health and the conditions that determine such perceptions. Knowing these dimensions and how they are interrelated, unifying criteria, helps to simplify asset mapping actions, differentiating these from other resources, and will facilitate the process for administrators to promote community interventions that improve health.

Moreover, it is also noteworthy that many of the studies reviewed, and some of the studies included in the content analysis, analysed personal-social assets separately from physical assets. This over-emphasis on individual and collective psychosocial resources and the accompanying silence concerning material assets has been criticised by authors such as Friedli [19] and O’Connor et al. [26].

In agreement with our review, McLean and McNeice (2012) as cited in Friedli [19] have observed that many asset mapping studies or projects are “case study” assessments, rendering it impossible to answer questions about the effectiveness of such interventions, because an asset-based approach to public health assumes certain inherent community circumstances that render more traditional assessment methods, such as the randomised control trial, less useful and at times inappropriate [20]. Similarly to our review, Agdal, Midtgard and Meidell [103] observed that the participatory action research (PAR) method is used as a foundation for the mapping process.

In line with the widespread interest in determining synergies between the salutogenic approach and the deficit approach to leverage the complementarity of both, recognising the dialectic links between needs and assets [14,16,17], our systematised review has enabled us to identify the needs that are most closely related to the mapped assets. The model proposed in this study can be used to assess resources based on the fundamental needs that render them meaningful. In this way, asset maps acquire a more dynamic condition that changes as the needs of the community do, and certain resources that were not previously health assets have the option and potential to be so. The result is 14 dimensions and 25 subcategories that form a guide to perceptual and objective considerations that the extensive literature in various disciplines (from social and environmental psychology to ecology, urban planning and economics) supports as meaningful aspects with health outcomes.

To the best of our knowledge, no previous study of health assets has attempted to identify the dimensions that characterise the salutogenic capacity of community resources. However, we did find reviews that examined categorical variables for specific environments or contexts, such as parks or other public spaces [104,105] and leisure resources [81]; or that were related to specific needs or dimensions such as accessibility [106], walkability [107], design [39], and sustainability [108]; or that were based on other related approaches, such as the study by Badland et al. [69], which concludes with a list of criteria for measuring the social determinants of health, and the study by Smith et al. [109] on “liveability”. The proposed model connects these variables of influence on health, and adds the focus of meaningfulness for the individual or the community.

## 5. Conclusions

Our study demonstrates the paucity of research specifically aimed at identifying the characteristics of health assets or determining the reasons why one asset or another is identified in the mapping process.

The dimensions identified by means of our systematised review were accompanied by in-depth, cross-disciplinary reviews of these concepts that enabled us to specify particular indicators or appropriate items for each of them.

Our proposal links Aaron Antonov sky’s salutogenic orientation (in the sense of manageability, comprehensibility and meaningfulness) with the health asset mapping approach, itself grounded in basic human needs and dimensions related to design, diversity, sustainability and other determinants. Moreover, it differs from previous studies in that it includes dimensions that refer to the meaningfulness of an asset for the individual (utility, intention, previous use, and even the dimension of identity).

Our proposal contributes to the necessary construction of a standardised method that adopts the salutogenic approach, oriented towards health equity and based on measurable and verifiable criteria. It is important to guide assessment not only of the results, but also of the process carried out, to ensure the principles of equity and “true participation” of the community.

## Figures and Tables

**Figure 1 ijerph-18-05758-f001:**
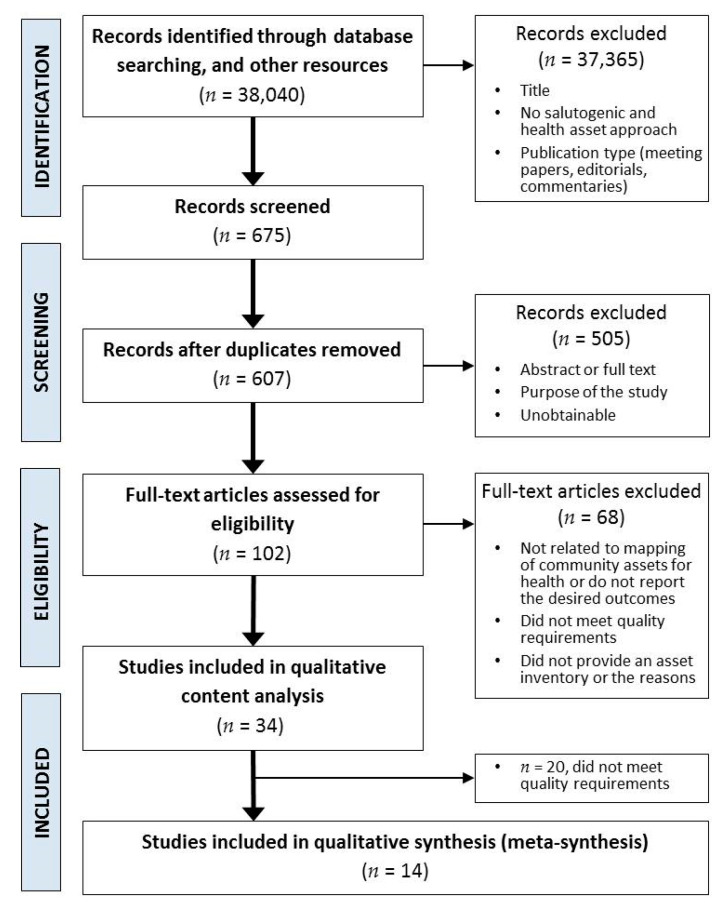
Adapted PRISMA flow diagram of the study selection.

**Table 1 ijerph-18-05758-t001:** General characteristics of the documents included. Meta-synthesis (*n* =14).

Reference Number	Authors, Date	Place and Context	Study Design	Approach	Study Population	Outcomes Inventory/Reasons	Quality Assessment
[26]	O’Connor et al. (2019)	Victoria (Australia), n/d.	Qualitative descriptive (focus groups and interviews)	AB ^1^	41 (university of the third age and primary school)	Mixed Yes/yes	++
[27]	Mosavel et al. (2018)	Petersburg (USA), 2012	PAR ^2^ (photovoice and GIS mapping)	CBPR ^3^	22 (young students and university students)	Only community Yes/yes	++
[28]	Sánchez-Casado et al. (2017)	Valencia city (Spain), May-July 2014	PAR(mapping workshops)	Salutogenesis; HA	29 (healthcare managers and professionals)	Mixed Yes/yes	+
[29]	Den Broeder et al. (2017)	Amsterdam (Netherlands), n/d.	Qualitative descriptive (nominal groups and interviews)	AB	21 (health professionals)	Only community Yes/yes	++
[10]	Aviñó (2017)	Valencia city (Spain), 2010	PAR (multi-method)	ABCD ^4^	Two case studies 106 (professionals and social fabric)	Mixed Yes/yes	++
[30]	Florian et al. (2016)	Massachusetts (USA), April 2015	PAR (photovoice and GIS mapping)	CBPR	17 (patients with diabetes)	Only community Yes/yes	++
[31]	Cutts et al. (2016)	Erijaville (South Africa) and Memphis (USA), n/d.	PAR (mapping workshops)	CBPR	Two case studies 100 (varied social fabric)	Mixed Yes/yes	++
[32]	Jabeen (2015)	Dhaka (Bangladesh), Sept.2010 to Mar.2011	Qualitative descriptive (focus groups and questionnaire)	AB	180 (dwellings)	Mixed Yes/yes	++
[33]	Pérez-Wilson et al. (2015)	Granada (Spain), Jun-Sept. 2011	Qualitative descriptive (focus groups and interviews)	Salutogenesis; HA	34 (adolescents, teachers and nurses)	Mixed Yes/yes	++
[11]	Jakes et al. (2015)	North Carolina (USA), 2012-2013	PAR (mapping workshops and interviews)	HA; CBPR	84 (varied social fabric)	Only community Yes/yes	++
[34]	Matthiesen et al. (2014)	Cumbria, Merseyside and Cheshire (England), 2011	PAR (mapping workshops and community event)	ABCD	Two case studies (93 professionals and 143 organisations)	Mixed Yes/yes	+
[35]	DyckFehderau et al. (2013)	Alberta (USA), Aug. 2008 to Oct. 2009	PAR (photovoice and discussion mapping)	CBPR	Students (11–16 years old)	Only community Yes/yes	+
[36]	Greetham et al. (2012)	Wakefield (England), 2010	PAR (multi-method)	ABCD	43 (varied social fabric)	Mixed Yes/yes	+
[37]	Lazarus et al. (2010)	Swellendam (South Africa) Feb-Nov. 2010	PAR (multi-method)	CBPR	295 (varied social fabric)	Mixed Yes/yes	+

^1^ AB = Asset-Based Approach. ^2^ PAR = Participatory Action Research (Kurt Lewin, 1946). ^3^ CBPR = Community-Based Participatory Research. ^4^ ABCD = Asset-Based Community Development [3]. HA = Health Asset Model [5].

**Table 2 ijerph-18-05758-t002:** Concepts and concurrences identified in the content analysis. ATLAS.ti.

Source	Link	Origin
Accessibility	is cause of	Walkability
Economic accessibility	is property of	Affordability
Adaptability	is property of	Sustainability
Appropriation	is part of	Identity
Affordability	is property of	Accessibility
Physical barriers	is cause of	Walkability
Community capital	is associated with	Sustainability
Features	is property of	Design
Comfort	is associated with	Design
Commitment	is property of	Participation
Confidence	is cause of	Safety
Time availability	is property of	Opportunity
Equity/inclusiveness	is property of	Intersectorality
Equity/inclusiveness	is associated with	Participation
Open spaces	is associated with	Public
Aesthetic	is cause of	Walkability
Aesthetic	is property of	Design
Strategic-reflective	is property of	Organisational structure
Organisational structure	is property of	Intersectorality
Utility	is associated with	Meaningfulness
Previous use	is property of	Utility
Attitude	is associated with	Meaningfulness
Funcionality	is associated with	Multifunctionality
Abilities	is associated with	Walkability
Illumination	is cause of	Safety
Information	is part of	Legibility
Intersectorality	is associated with	Participation
Intersectorality	is property of	Sustainability
Maintenance	is associated with	Features
Maintenance	is cause of	Safety
Fresh/nature	is associated with	Peace/calm
Opportunity	is property of	Affordability
Participation	is cause of	Safety
Participation	is associated with	Socialisation
Participation	is cause of	Abilities
Participation	is cause of	Manageability/control
Participation	is cause of	Meaningfulness
Participation	is associated with	Identity
Stable/durable	is property of	Sustainability
Proximity	is cause of	Walkability
Safety	is cause of	Walkability
Security	is part of	Safety
Socialisation	is associated with	Safety
Socialisation	is cause of	Abilities
Adaptability	is property of	Sustainability
Mixed land uses	is cause of	Walkability
Variety (offer/service)	is associated with	Funcionality
Variety (offer/service)	noname	Multifunctionality

**Table 3 ijerph-18-05758-t003:** Dimensions and categories of community assets for health.

Dimension	Categories	Concepts
Utility	-	Fundamental needs
Intention (personal)	Subjective Norm	-
	Attitude	-
	Motivation and desire	-
Previous use	-	-
Affordability	Circumstances Opportunity Economic accessibility	-
		Time
		-
Proximity	-	-
Walkability	- -	Rectitude
		Integrity
Connectivity	-	-
Legibility	Visibility Transparency/clarity	-
		-
Identity	Singularity	-
	Appropriability	-
	Attachment	-
Design	Configuration	Features/Characteristics
		Aesthetic
	Funcionality	Flexibility
		Multifuncionality
	Comfort	-
Safety	Security (perceived)	-
	Security (objetive)	-
Diversity	-	Quantity
	-	Variety
Public	Public	-
	Privacy	-
Sustainability	Durability	Maintenance
	Economic sustainability	Social cost effectiveness
	Environmental sustainability	-
	Centrality	Participation
		Betweenness centrality
		Closeness
	Equity	-
	Inclusiveness	-

## Data Availability

The data is presented in the paper.

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
