# Peer review of "Dimensions of Community Assets for Health. A Systematised Review and Meta-Synthesis"

_ijerph, 2021, doi:10.3390/ijerph18115758_

Round 1
Reviewer 1 Report
Dear authors,
I am honored to review this manuscript which is aimed to providing a comprehensive and systematic review of 14 qualified literature on health asset mapping. Finally, 14 dimensions and 12 subcategories were identified after solid meta-analysis.
This is a good review article and my only concern is the length of the content is too long that may reduce the value and readers' interests to this interesting paper. So, I would like to recommend the authors to concise the paper as possible without sacrifice of its original contents.
Author Response
This is a good review article and my only concern is the length of the content is too long that may reduce the value and readers' interests to this interesting paper. So, I would like to recommend the authors to concise the paper as possible without sacrifice of its original contents.
Thanks for your comments.
The length of the article has been reduced to the maximum possible, considering not to lose any valuable information for its understanding. For example, the phrases corresponding to lines 81-83 , 93-97, 168-170 or 316-320 of the original document, before the modifications.
Reviewer 2 Report
Overall, this paper is well written with a solid methodological approach. It covers important information and can be helpful to researchers and community activists to better understand assets and how they may be helpful going forward in improving community health outcomes. Where the paper fall short is an explicitly identifying its purpose and carrying that common theme throughout the paper. The research aim specifically identifies the development of the tool as its purpose. If that is the case, the Discussion should be rewritten to focus on the benefits of this tool and how it may be implemented into the future. If the purpose is as it is written on line 560 at the beginning of the Discussion, then I would suggest tightening the Discussion to reflect how this tool can be used to first map community assets and then how to use that mapping within a health promotion framework to be able to improve community health outcomes. Further, the Conclusion introduces the concept of health equity; however, this concept is only introduced at the beginning of the paper in relation to the social determinants of health it does not seem to take a prominent role throughout the rest of the paper. If the intention of the authors is to use their tool to improve health equity, it should become a focus of the Discussion. Specific Comments: The term "in order" is used numerous times throughout the paper. It can be removed in almost all instances. Line 99 - the listing of the questions here is curious. I understand the motivation to include those as they guided your research, but they are not addressed in the paper. If these questions remain included, you may wish to return to them in your conclusion. Figure 1 is typically referred to as a PRISMA Flow Diagram. A little more detail on how you categorize good and poor quality papers and therefore excluded some might be helpful. The Conclusion does not answer the purpose of the paper and should rewritten to do so.Author Response
Thanks a lot for your comments!
The term "in order" is used numerous times throughout the paper. It can be removed in almost all instances.
Modified in most occasions.
Line 99 - the listing of the questions here is curious. I understand the motivation to include those as they guided your research, but they are not addressed in the paper. If these questions remain included, you may wish to return to them in your conclusión.
We consider this contribution very appropriate, which is why we have considered it in the modification of the conclusions.
Figure 1 is typically referred to as a PRISMA Flow Diagram.
Modified
A little more detail on how you categorize good and poor quality papers and therefore excluded some might be helpful.
Done
The research aim specifically identifies the development of the tool as its purpose. If that is the case, the Discussion should be rewritten to focus on the benefits of this tool and how it may be implemented into the future. If the purpose is as it is written on line 560 at the beginning of the Discussion, then I would suggest tightening the Discussion to reflect how this tool can be used to first map community assets and then how to use that mapping within a health promotion framework to be able to improve community health outcomes.
The aim of this study is the identification of the characteristics or dimensions that differentiate health assets. This formed a model and an initial tool that, in a second moment, was subjected to its validation. For this reason, the discussion has been modified in order to make the importance of this new model more evident to readers.
Further, the Conclusion introduces the concept of health equity; however, this concept is only introduced at the beginning of the paper in relation to the social determinants of health it does not seem to take a prominent role throughout the rest of the paper. If the intention of the authors is to use their tool to improve health equity, it should become a focus of the Discussion.
Equity is one more category like others mentioned, particularly it belongs to the dimension of Sustainability. We modified the conclusions so as not to highlight it over the others, although this variable had greater weight in the subsequent validation of the tool.
Round 2
Reviewer 2 Report
None.